# VoCap: Video Object Captioning and Segmentation from Any Prompt

## Abstract

Understanding objects in videos in terms of fine-grained localization masks and detailed semantic properties is a fundamental task in video understanding. In this paper, we propose VoCap, a flexible video model that consumes a video and a prompt of various modalities (text, box or mask), and produces a spatio-temporal masklet with a corresponding object-centric caption. As such our model addresses simultaneously the tasks of promptable video object segmentation, referring expression segmentation, and object captioning. Since obtaining data for this task is tedious and expensive, we propose to annotate an existing large-scale segmentation dataset (SAV) with pseudo object captions. We do so by preprocessing videos with their ground-truth masks to highlight the object of interest and feed this to a large Vision Language Model (VLM). For an unbiased evaluation, we collect manual annotations on the validation set. We call the resulting dataset SAV-Caption. We train our VoCap model at scale on a SAV-Caption together with a mix of other image and video datasets. Our model establishes a benchmark for video object captioning and yields state-of-the-art results on referring expression video object segmentation. Our dataset will be made available.

## 1 Introduction

Understanding objects in videos, including both their fine-grained locations (represented as segmentation masks) as well as their detailed semantic properties, is a fundamental task in video understanding. It serves as a basic block for various applications, including video generation and editing (Chai et al., 2023; Hu et al., 2024; Wang et al., 2024b), wildlife conservation (Beery et al., 2020; Sun et al., 2024), and self-driving cars (Caesar et al., 2020; Sun et al., 2020). While it is trivial for a human to point to an object in a video and describe it in detail, there is yet no existing computer vision system that is capable of both *spatio-temporal* localization via segmentation masks, as well as a *semantic* understanding of objects via natural language.

In this paper, we propose a model and data for fine-grained video object understanding with flexible inputs and outputs modalities. Our model consumes a video and an input prompt, where the prompt can be a mask and box, but also natural language (i.e. referring expression). Our model then produces *both* a spatio-temporal mask (i.e., a 'masklet') *and* a free-form natural language caption describing the object. Because the output caption is a free-form sentence, it can describe the attributes of the object as well as how they change over time. Our model can be used for a variety of tasks bridging localization and language, for example referring object segmentation (Yu et al., 2016; Seo et al., 2020) or location-conditioned captioning (Krishna et al., 2017b), which we extend to video.

Several previous works attempt to bridge this gap between visual localization and language understanding: in segmentation via free-form referring expressions (Cuttano et al., 2025; Khoreva et al., 2018; Seo et al., 2020; Wu et al., 2022) the goal is to produce a segmentation mask for an object given a short description which refers to a single object. The task of dense video object captioning (DenseVOC) (Zhou et al., 2023) aims to produce bounding boxes and captions for all classes within a certain vocabulary in a video. While localization with referring expressions (Cuttano et al., 2025; Khoreva et al., 2018; Seo et al., 2020; Wu et al., 2022) typically only takes in a minimal-required text to identify an object in the input, our model can also produce detailed captions given a location prompt. Unlike DenseVOC (Zhou et al., 2023) – which is non-promptable, is trained on a fixed set of objects, and which is limited to producing boxes only – our model works with flexible input prompts

Figure 1: **Overview of our VoCap architecture.** Our model processes videos frame-by-frame, with access to an updating memory for each object. Each frame goes through the image encoder, cross attends to the memory. The memory-aggregated image features and the object-specific prompt embeddings are fed into the mask decoder to obtain the mask predictions. The memory module is updated with the per-frame mask predictions and image features. In addition, our model also takes text prompts, and we use a novel text feature extractor and text decoder to produce captions for the object. The text encoder and text decoder share the architecture and weights.

and produces dense masklets as output in addition to captions. Our model is inspired by both existing captioning Vision-language models (VLMs) such as BLIP2 (Li et al., 2023), PaliGemma2 Steiner et al. (2024), and Qwen-VL Bai et al. (2023), as well as promptable segmentation models such as SAM2 (Ravi et al., 2024), and brings a number of related video segmentation and captioning tasks together while enabling cross-task synergies. Specifically, we augment the general SAM2 design (Ravi et al., 2024) by adding caption tokens, a text feature extractor and a text decoder to enable captioning in the spirit of QFormer (Li et al., 2023), and by adding a text encoder to the input prompt. Our design enables sharing weights between the text encoder and decoder to unlock synergies between referring expression inputs and captioning output. This results in a unified promptable model for Video Object Captioning and Segmentation from Any Prompt (VoCap) that takes as input a prompt (text, mask, or box) and outputs segmentation masks and a caption jointly (See Fig. 1).

Obtaining data to train our VoCap model is a significant challenge – annotating video with segmentation masklets and captions is tedious and expensive, and not easily scalable to large volumes of data. Hence we propose a pseudo-labeling pipeline starting with the SAV Manual dataset (Ravi et al., 2024), which contains accurate segmentation masks. We then automatically generate *object-centric* captions using a large-scale VLM (Gemini 1.5 Pro Vision (Gemini Team, 2024)). By pre-processing the videos to highlight each object mask and blur the background, we steer the VLM to describe each object and what happens to it with satisfying accuracy and details. This enables us to generate a large-scale training set with masks and object-centric captions without additional human labor. We then combine this dataset with existing datasets (Krishna et al., 2017b; Seo et al., 2020; Yu et al., 2016; Chen et al., 2022) to co-train our model. We note that each of these datasets cover only a subset of our input and output modalities.

For evaluation, we ran a *human* annotation campaign on the SAV-val dataset (Ravi et al., 2024) where each object is captioned by three different annotators. Furthermore, we evaluate our model on existing referring expression video object segmentation datasets and image captioning. To summarize, we make the following contributions:

- We present VoCap, a unified promptable model that can produce both spatio-temporal masklets and captions for objects in video. Our model is flexible in both the input and the output, taking as input a prompt (text, mask, or box) and outputting masklets and captions. Our model is the first to support all these input and output modalities. We demonstrate synergies between the modalities; e.g. training the model for captioning (language at output) improves its referring expression comprehension (language at input).

- We collect manually annotated object captions on SAV-val and create pseudo-captions by leveraging existing mask annotations on SAV-train using Gemini Pro 1.5. We will make both the manual and pseudo-annotations publicly available By showing good performance on the manually annotated object captions, we demonstrate that these pseudo-labels are effective for training our captioning model.

- We establish a benchmark for video object captioning and set a new state-of-the art for Referring Expression Video Object Segmentation.

## 2  RELATED WORK

**Segmentation and Captioning Models.** A variety of models are dedicated to video segmentation and expect an initial input mask (Yang et al., 2021; 2024; Yang & Yang, 2022; Cheng & Schwing, 2022; Cheng et al., 2024; Guo et al., 2024; Deng et al., 2024; Ravi et al., 2024; Yang et al., 2023c), a referring expression (Seo et al., 2020; Lan et al., 2023; Wu et al., 2022), or both (Cheng et al., 2023b; Cuttano et al., 2025; Wu et al., 2023). CLIPSeg (Lüddecke & Ecker, 2022) can also consume a query image. SAM2 (Ravi et al., 2024) can do segmentation without any inputs as is done in DEVA (Cheng et al., 2023b) by starting from the original SAM (Kirillov et al., 2023) with a point grid prompts. However, none of these models can generate descriptions. In captioning, there are works on *global video* captioning which describe the whole video (Kanani et al., 2021; Iashin & Rahtu, 2020; Yang et al., 2023a; Yao et al., 2015; Wang et al., 2021a), or on *dense image* captioning which provide object-centric captions and their locations in images (Johnson et al., 2016; Li et al., 2019; Shao et al., 2022; Zhang et al., 2023; Yuan et al., 2024; Peng et al., 2023; Xu et al., 2024). Only few works do *dense video* captioning (Choudhuri et al., 2024; Zhou et al., 2023) for objects. DenseVOC Zhou et al. (2023) predicts bounding boxes with captions but does not predict masks and only detects a predefined set of object classes. The OW-VISCapTor model (Choudhuri et al., 2024) predicts segments with captions, but cannot handle textual or mask input prompts. Furthermore, OW-VISCapTor is based on an image-first *tracking-by-detection* paradigm that can be suboptimal in long videos with occlusions, while we build on top of strong *memory-based* trackers (Ravi et al., 2024; Cheng & Schwing, 2022) and can handle long and challenging videos (Ding et al., 2023b).

**Datasets.** While numerous datasets exist for video object segmentation and captioning separately, very few combine both on the same set. Video segmentation datasets include various input forms, including a mask given on the first frame (*semi-supervised video object segmentation*, or *SS-VOS*) (Perazzi et al., 2016; Caelles et al., 2019; Ding et al., 2023b; Qi et al., 2022; Wang et al., 2021b), a target object class (*semantic object segmentation*) (Kim et al., 2020; Real et al., 2017; Russakovsky et al., 2015) or a referring expression (Ding et al., 2023a; Khoreva et al., 2018; Seo et al., 2020; Wu et al., 2022) (*Referring Video Object Segmentation*, or *RefVOS*). While referring expression datasets have masks and referring text, this text tends to be mainly focused on identifying the object in the video, not describing it. To link captions better to the visual domain, several datasets focus on having *grounded* captions in both the image domain (Krishna et al., 2017b; Lin et al., 2024; Plummer et al., 2015; Peng et al., 2023; Pont-Tuset et al., 2020; Wang et al., 2023b; 2024a; Xue et al., 2024) and video domain (Voigtlaender et al., 2023; Zhang et al., 2020; Zhou et al., 2018a; 2019). In particular, in the video domain, bounding box annotations are added in (Zhou et al., 2018a) to YouCook2 (Zhou et al., 2018b), and in (Zhou et al., 2019) to ActivityNet (Krishna et al., 2017a). In (Zhang et al., 2020) the relations of VidOR (Shang et al., 2019) are converted into captions while grounding is provided by the existing bounding boxes. BenSMOT (Li et al., 2024) provides a human-focused dataset with boxes, their object-centric captions, and interactions. Video Localized Narratives (Voigtlaender et al., 2023) introduced an object-centric protocol in which captions are grounded by a mouse trace. In contrast, in this paper we provide a stronger form of grounding by linking captions to *segmentation masks*. Our pseudo-labeled dataset is also an order of magnitude larger than these datasets (Tab 1).

**Pseudo labels.** With increasing model capabilities and increasing data requirements for training large models, it is increasingly common to use automatically generated labels in the pre-training stage. SAM2 (Ravi et al., 2024) provides automatically generated masks on their SAV dataset, enabling distillation. BLIP3 (Xue et al., 2024), OWLv2 (Minderer et al., 2023), and Kosmos-2 (Peng et al., 2023) go beyond distillation for bounding box generation by exploiting existing captions: they extract noun phrases from the captions, feed them to an open-world detector, and only keep high-scored boxes. MVDP (Lin et al., 2024) draws existing object classes and location annotations in an image using set-of-masks (Yang et al., 2023b) and feeds this to GPT-4V (OpenAI, 2023,) to generate object-centric captions, relationships, and Q&A pairs. In this paper, we augment videos with ground-truth segmentation masks and prompt vision-language models to create high-quality pseudo captions.

## 3  THE SAV-CAPTION DATASET

We want to have a large-scale training set with spatio-temporal segmentation masks and their captions. Therefore we start from SAV (Ravi et al., 2024), the largest and most diverse video dataset

Table 1: **Video datasets with masks and captions.** Our SAV-Caption is an order of magnitude larger.

| dataset | # videos | # objects captioned | # words per caption |
|---|---|---|---|
| RefVOS-DAVIS (Khoreva et al., 2018) | 150 | 436 | 6.4 |
| MeVIS (Ding et al., 2023a) | 2.0k | 8.1k | 7.3 |
| RefVOS-YTVOS (Seo et al., 2020) | 4.0k | 7.5k | 9.7 |
| SAV-Caption val (manual) | 155 | 290 | 13.5 |
| SAV-Caption train (automatic) | 50k | 170k | 11.8 |

with segmentation masks. We use the 'Manual' part which was annotated by combining SAM2 predictions (Ravi et al., 2024) with human annotator corrections to ensure high-quality masklets. Next we detail how we add captions to the existing SAV segmentation dataset using automatic annotations and human annotations. In both cases we want to have captions with the object class, its visual properties (which aligns the captions with visual referring expressions), and what it does (which captures the temporal semantics). Such captions are aligned with previous dense video captioning datasets (e.g. Voigtlaender et al. (2023); Zhang et al. (2020)).

**Automatically Annotated Training Data**   We use Gemini 1.5 Pro Vision (Gemini Team, 2024) to automatically generate captions on this dataset. This model is a long-context vision language model and is therefore suited to consume relatively large video clips ($\approx 1000$ frames). To create accurate captions, we draw inspiration from works which augment images with visual prompts to focus the attention of the visual models to what matters, thereby simplifying the task (Nasiriany et al., 2024; Yang et al., 2023b; Zheng et al., 2024; Wu et al., 2024c; Shtedritski et al., 2023). In particular, we adopt two visual prompting techniques: 1) We highlight the target segment by drawing a clear red contour around it (**Contour**); 2) upon finding that Gemini would still sometimes focus on objects in the background, we blurred the background using a Gaussian filter (**Blur**). Both modifications are explicitly mentioned in the textual prompt. An example of the video frame we fed to Gemini can be seen in the Appendix in Fig. 3.

For the textual part of the prompt, we carefully iterated to increase the quality of the generated caption. In this process, we found it helpful to structure the prompt: we ask to describe first the object, then its visual properties, and then what it does, and finally we ask it to give the caption while keeping earlier mentioned elements consistent. Statistics of SAV-Caption train are given in Tab. 1, and example captions are shown in Fig. 4 of the Appendix. A quantitative analysis of the quality of the generated captions together with the exact prompt used is given in Appendix A.

**Human Annotated Validation Data**   Our evaluation should be free of any potential biases of any Visual Language Model. Therefore we collect our evaluation set fully manually with three captions per object. In particular, we start from SAV-val and instruct the raters to provide a single free-form caption of the object highlighted in the video. Like in Sec. 3 we highlight the object with a red border but we do not blur the background. We have explicit instructions for the annotators to include in their caption the object class, its visual properties, and what it does. We also ask them to not mention irrelevant objects in the background. The statistics of SAV-Caption val are given in Tab. 1. The annotation instructions and the UI can be found as a separate file in the supplementary material.

**Comparison with Other Datasets**   There only exist few video datasets where objects are annotated with both spatio-temporal masklets and captions (Ding et al., 2023a; Khoreva et al., 2018; Seo et al., 2020). These existing datasets were all made for referring expression segmentation but can be repurposed for the captioning task. However, referring expressions were made with the intention for objects to be uniquely identifiable, not for semantic understanding. Furthermore, our training set is at least one order of magnitude bigger.

## 4 VOCAP MODEL

Given an image or video $V \in \mathbb{R}^{T \times H \times W \times 3}$ (for images, $T = 1$) and a prompt, where the prompt can be a bounding box or a mask in the first frame, or a textual description, our VoCap model produces a binary masklet $M \in \mathbb{R}^{T \times H \times W}$ and a caption string $\mathbf{s}$ for the corresponding object.

## 4.1 MODEL ARCHITECTURE

As illustrated in Fig. 1, our model is composed of segmentation modules inspired by (Ravi et al., 2024), including an image encoder, a memory encoder, a memory attention module, a location prompt encoder, and a mask decoder. We add new language modules: a text encoder, a text feature extractor, and a text decoder. As a result, our model can take both texts or masks as inputs or as outputs.

The **image encoder** takes a single frame $V_t$ as input and produces down-sampled image features $\mathbf{f}_t \in \mathbb{R}^{H' \times W' \times d}$. This can be any visual backbone, and we use EVA02-L Fang et al. (2023) given its dedicated pretraining for both language (Radford et al., 2021) and localization tasks (He et al., 2022). Following ViTDet (Li et al., 2022), we use simple convolutional upsampling layers (Ronneberger et al., 2015; Zheng et al., 2021) to produce multi-scale features as additional inputs for the mask decoder (Ravi et al., 2024). Note that each frame is processed separately without temporal communication.

The **memory encoder** and **memory attention** together augment the per-frame image feature $\mathbf{f}_t$ with temporal information. Specifically, at each timestamp, the memory encoder fuses the input image and output mask into a memory feature, which is stored in a memory bank that keeps a history of $d'$-dimensional spatio-temporal appearance features (memory dimension $d'$ can be different from feature dimension $d$). Following SAM2 (Ravi et al., 2024) we use a fixed-sized memory bank with a first-in-first-out memory queue. There are several cross attention layers between the current image feature and the memory bank which makes the output image features $\bar{\mathbf{f}}_t \in \mathbb{R}^{H' \times W' \times d}$ temporally-aware.

The **location prompt encoder** projects location inputs to embeddings. Specifically, box prompts are encoded as sparse embeddings $\mathbf{p} \in \mathbb{R}^{n \times d}$, where $n$ is the number of points ($n = 2$ for 2 box-corners) and $d$ is the feature dimension. Mask prompts are encoded as dense embeddings $\mathbf{m} \in \mathbb{R}^{H' \times W' \times d}$ with the same shape as the image feature.

The **text encoder** takes text strings as inputs and projects them to embeddings. It can be any language model (Devlin et al., 2019; Team et al., 2024a;b; Raffel et al., 2020; Touvron et al., 2023) that encodes the integer vocabulary indexes to embeddings. Specifically, we feed text prompts as the text prefix to the language model with full attention, and extract the features before the vocabulary classification layer. We use an additional dimension-matching layer to project from the language model embedding space to the prompt embedding space. We reuse our sparse embedding notation $\mathbf{p} \in \mathbb{R}^{n \times d}$ for text prompts. Here $n$ is the number of tokenized words in the text query. Because the text prompt provides conditioning for the entire video and because the target object does not always appear in the early frames of the video, we feed the text prompt embedding to all frames of the video.

The **mask decoder** takes the temporal-aware image feature $\bar{\mathbf{f}}_t$ and the prompt features $\mathbf{p}$ or $\mathbf{m}$ as inputs, and outputs the mask at the current frame $M_t \in \mathbb{R}^{H \times W}$. In SAM, the mask decoder uses cross attention to communicate the image and prompt features:

$$\tilde{\mathbf{f}}_t, [\tilde{\mathbf{p}}, \tilde{\mathbf{o}}] = CA_{seg}(\bar{\mathbf{f}}_t + \mathbf{m}, [\mathbf{p}, \mathbf{o}]) \tag{1}$$

$$m_t = \mathcal{D}(\tilde{\mathbf{f}}_t, \tilde{\mathbf{o}}) \tag{2}$$

where $\mathbf{o} \in \mathbb{R}^{1 \times d}$ is a learned mask token and is concatenated with the sparse prompt $\mathbf{p}$, and $CA$ is the cross-attention operation. $\mathcal{D}$ is a mask decoding function with upsampling convolutions and a final dot-product (Cheng et al., 2021). The output of the cross-attention, $\tilde{\mathbf{o}} \in \mathbb{R}^{H \times W}$, can be considered as the object feature conditioned on the prompt. Besides the mask, the mask decoder also predicts for each frame a binary object appearance indicator $a_t \in \{0, 1\}$ to handle occlusion or out-of-view movement, and an IoU prediction $iou_t$ which estimates the quality of the mask.

**Text feature extractor**. Similar to how the object features are extracted in the mask decoder in Eq. 1, we use learned caption tokens $\mathbf{c} \in \mathbb{R}^{l \times d}$ and cross attention to extract caption features for each object:

$$\hat{\mathbf{f}}, [\hat{\mathbf{p}}, \hat{\mathbf{c}}] = CA_{cap}(\bar{\mathbf{f}}_t + \mathbf{m}, [\mathbf{p}, \mathbf{c}]) \tag{3}$$

where we only use output $\hat{\mathbf{c}}$ and discard $\hat{\mathbf{f}}$ and $\hat{\mathbf{p}}$. This formulation is analogous to popular vision-feature extractors in vision-language models (Jaegle et al., 2021; Alayrac et al., 2022; Ryoo et al., 2021; Li et al., 2023), while we additionally condition on the prompt embeddings $\mathbf{m}$ or $\mathbf{p}$. Following BLIP2 (Li et al., 2023), we use $l = 32$ tokens for the caption tokens.

**Text decoder**. Following popular vision-language model design (Li et al., 2023; Wang et al., 2022; Liu et al., 2023) we feed the object-aware caption feature $\hat{\mathbf{c}}$ as prefix to an auto-regressive language model $\mathcal{L}$ to produce object caption $\mathbf{s}$:

$$\mathbf{s}_i = \mathcal{L}(\hat{\mathbf{c}}, \mathbf{s}_{1:i-1}) \tag{4}$$

Again, the text decoder can be any language model (Devlin et al., 2019; Team et al., 2024a;b; Raffel et al., 2020; Touvron et al., 2023) with a causal attention mask. We note that both the architecture and the weights of the text encoder and text decoder can be shared even though the text decoder uses causal attention, and the text encoder uses bidirectional attention. Therefore, during training, the language model is updated for both text encoding and decoding regardless of whether we use text as an input prompt or as a target output caption. We follow the standard transformer decoder (Vaswani et al., 2017) as it is simple and effective (Wang et al., 2022; Wu et al., 2024a; Zhou et al., 2023).

For some more implementation details we refer the reader to Appendix B.

### 4.2 TRAINING

Given our flexibility on inputs and outputs, our model can leverage a variety types of annotations from different datasets: For SAV-Caption our model consumes a mask prompt and calculates the loss on both the predicted masklet and the predicted caption. On VisualGenome (Krishna et al., 2017a) our model consumes a box prompt and calculates the loss on the caption. For SS-VOS we have a first frame mask input prompt and a loss on the masklet. For RefVOS we have a text input prompt and a loss on the masklet. Following other joint models for image and video, we treat images as a single-frame video (Ravi et al., 2024; Villegas et al., 2022; Bain et al., 2021). Concretely, we do not use the memory module (specifically, for $t = 0$, $\bar{\mathbf{f}}_0 \equiv \mathbf{f}_0$) for the first frame or images. To leverage all available data, we first pre-train our language and vision components separately, then perform multi-task training with joint mask- and caption-annotations. Finally, for achieving the best performance, we finetune on specific datasets. See more details in Appendix B and D.

### 4.3 INFERENCE

Our model runs on images or videos of arbitrary lengths. Like in training, for an image or the first frame of the video, the visual features are not modified by the memory attention since there are no memories (again, $\bar{\mathbf{f}}_0 \equiv \mathbf{f}_0$). For the following frames of the video, our model runs in an online manner: in each frame the model produces both mask and caption outputs, and updates the memory. For the final caption prediction we take the one from the last frame; through the memory this caption is conditioned on the previous frames and therefore captures the temporal aspect of the video.

## 5 EXPERIMENTS

### 5.1 CAPTIONING

The localized captioning task is defined as producing a text caption given a location prompt (e.g. box or mask). We are the first to propose this task for video, where we aim to produce both a caption and a spatiotemporal segmentation given a mask annotation for the first video frame. Since our method also works on images, we evaluate on image captioning on Visual Genome given a location prompt in the form of a box around the object. This enables us to compare to state-of-the-art object captioning methods on images. For both captioning tasks we measure standard CIDEr (Vedantam et al., 2015).

**Video Object Captioning Baselines.** Since our VoCap model is the first model which can do simultaneous object segmentation and captioning given a first-frame input mask, there are no existing methods to compare to. Instead we present results for a few strong baselines. First, we run a semi-supervised VOS method to obtain segments, and feed these into existing off-the-shelf captioning models. In particular, we run our re-implemented and retrained SAM2 model (Ravi et al., 2024) as the SS-VOS method and apply the popular captioning models BLIP2 (Li et al., 2023) (which predicts captions from single images without any additional prompt) and PixelLLM (Xu et al., 2024) (which predicts captions from bounding-box location prompts in single images). For BLIP2 (Li et al., 2023), we follow CaptionAnything (Wang et al., 2023a) to use the SAM2 mask to crop and mask-out the background. For PixelLLM (Xu et al., 2024), we extract the bounding box as the prompt from the

Table 2: **Video Object Captioning Results.** We significantly outperform strong baselines in video object captioning. † The SAM2 numbers are from our retrained model as detailed in Appendix C.

| method | SAV-Caption-val (manual) | |
| --- | --- | --- |
| | captioning CIDEr | segmentation J&F |
| SAM2 † | ✗ | 75.8 |
| SAM2 † → BLIP2 (Li et al., 2023) | 21.9 | 75.8 |
| SAM2 † → PixelLLM (Xu et al., 2024) | 35.5 | 75.8 |
| SAM2 † → Gemini pseudo-labeling | 40.5 | 75.8 |
| UniRef++ → Gemini pseudo-labeling | 34.3 | 46.9 |
| VoCap (ours) | 47.8 | 75.5 |

Table 3: **Results on Localized Image Captioning.** The input is a box, the output a caption. We outperform all existing works.

| method | Visual Genome captioning - CIDEr |
| --- | --- |
| GRiT (Wu et al., 2024a) | 142 |
| PixelLLM (Xu et al., 2024) | 149 |
| SCA (Huang et al., 2024) | 150 |
| VoCap (ours) | 163 |

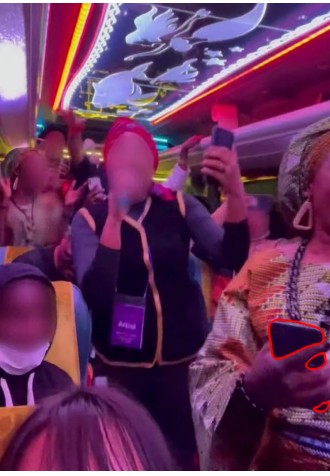 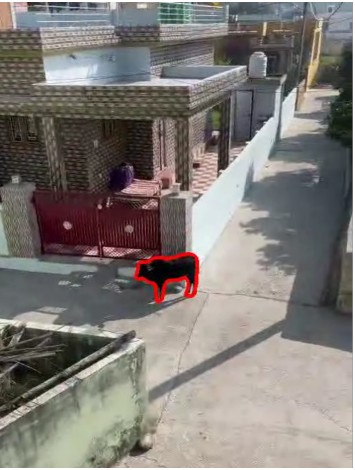

**GT:** A mobile phone, held in the hand of the person who is dancing.
**SAM2+Gemini:** A hand is holding something.
**VoCap:** A small, rectangular, dark phone is being held.

**GT:** A black cow is walking on the road near a house.
**SAM2+Gemini:** A small black dog is walking on the street.
**VoCap:** A black cow is walking.

Figure 2: Qualitative examples which illustrate where VoCap succeeds where SAM2+Gemini pseudo labeling does not. This typically happens in cases with small objects (both examples) and when an actor is nearby (the phone is held by a human hand).

SAM2 mask. These image baselines produce a caption in each frame, and we take a single video-level caption by taking the most common captions for the image caption sequence.

In addition, we create two baselines which closely follow our annotation pipeline: We use SAM2 (Ravi et al., 2024) and UniRef++ (Wu et al., 2023) to generate segmentation masks based on a first frame input mask. We feed these generated segments to our pseudo-annotation pipeline (Sec. 3).

**Video Object Captioning Results.** We finetune VoCap jointly on SAV-Caption-train and VisualGenome (Krishna et al., 2017b). Tab. 2 presents results on the SAV-Caption-val which was manually annotated (Sec. 3). VoCap significantly outperforms all baselines in captioning at only a minor decrease in segmentation performance compared to SAM2. In particular, BLIP2 (Li et al., 2023) and PixelLLM (Xu et al., 2024) yield suboptimal performance, likely since these image-based models do not capture motion. More importantly, our results (47.8 CIDEr) surpass applying SAM2 plus Gemini pseudo-labeling (40.5 CIDEr) despite being significantly more efficient (Gemini is much larger than VoCap). To understand how our model could outperform this strong baseline, we visually inspected the results. We observed that Gemini typically makes mistakes in small objects (Fig. 2, presumably due to resolution) and that it has an 'actor bias': it sometimes describes a human (hand) or animal which is near the highlighted object (Fig. 2, left). In contrast, since our model actively tracks an object, it always describes the object which it is tracking and not any object close by. From a more general learning perspective, by training on large amounts of data our model can correct or smooth out some of the noise of the pseudo-labels, which is a commonly observed phenomenon (e.g. Jia et al. (2021); Lee (2013); Radford et al. (2021)).

Table 4: **State-of-the-art comparison on Referring Video Object Segmentation (RefVOS).** We report official $J\&F$ metrics on each dataset. '-' means the paper does not report results. VoCap outperforms the state-of-the-art on all datasets.

| Dataset | RefVOS-DAVIS | RefVOS-YTVOS | MeViS | UVO-VLN |
|---|---|---|---|---|
| Point-VOS (Zulfikar et al., 2024) | - | - | - | 52.8 |
| ReferFormer (Wu et al., 2022) | 61.1 | 64.9 | - | 46.4 |
| SOC (Luo et al., 2024) | 67.2 | 67.3 | - | - |
| DsHmp (He & Ding, 2024) | 64.9 | 67.1 | 46.4 | - |
| FindTrack (Cho et al., 2025) | 74.2 | 70.3 | 48.2 | - |
| UniRef++ (Wu et al., 2023) | 67.2 | 67.4 | - | - |
| GLEE (Wu et al., 2024b) | - | 70.6 | - | - |
| SAMWISE (Cuttano et al., 2025) | 70.6 | 69.2 | 49.5 | - |
| VoCap (ours) | **75.1** | 70.3 | 51.9 | 62.2 |
| VoCap + FindTrack (ours) | 74.7 | **71.2** | **53.0** | 62.7 |

**Image Object Captioning.** There are several works on localized image captioning, where the input is an image and a bounding box around an object, and the output is the caption describing the object. Since our model can also consume box prompts, and since images can be interpreted as single-frame videos, we can directly compare to these works. We evalate the same VoCap model as before (finetuned jointly on SAV-Caption-train and VisualGenome) and evaluate it on the 5k validation images of VisualGenome (Krishna et al., 2017b) which has human-annotated object captions. Again, we report the standard captioning metric, CIDEr Vedantam et al. (2015). Results in Tab. 3 show that our method outperforms the state-of-the-art on this task: 150 CIDEr for SCA (Huang et al., 2024) vs 163 CIDEr for our VoCap model.

## 5.2 REFERRING EXPRESSION VIDEO OBJECT SEGMENTATION

For Referring Expression Video Object Segmentation (RefVOS) the input is a video and a textual referring expression of the target object. The output is a spatio-temporal masklet throughout the whole video which segments the object in every single frame.

**Datasets.** We evaluate on the popular video referring segmentation datasets RefVOS-YTVOS (Seo et al., 2020), RefVOS-DAVIS Khoreva et al. (2018), MeVis (Ding et al., 2023a) and UVO-VLN (Voigtlaender et al., 2023). For RefVOS-DAVIS (Khoreva et al., 2018) we follow UniRef++ (Wu et al., 2023) to only use its validation set of 30 videos as a zero-shot evaluation (on average 2 objects per video and with 4 text queries per object). The UVO-VLN Video Narrative Grounding (VNG) benchmark provides image descriptions and segmentation masks of labeled noun phrases. To turn a description into a referring expression (which should unambiguously refer to a single object) we simply mark the target noun with brackets (e.g. 'the dog catches the [frisbee]').

**FindTrack.** Now one problem with referring expressions is that the first frame may not have the clearest view of the object, it could be ambiguous (e.g. for 'the bird flying away' there could be three birds where one of them flies away only at the end), or not even visible at the first frame. Such cases can pose problems to our model since memory-based, online streaming segmentation models have been observed to be biased to keep tracking the object predicted in the first frame (Cho et al., 2025; Cuttano et al., 2025). To overcome this we also implemented the test-time inference method of FindTrack (Cho et al., 2025): We apply VoCap to each frame $t$ independently to produce masks with IoU predictions $iou_t$. We start from the mask and frame with the highest IoU prediction and from there we go both forward and backward in the video to produce a full masklet. Note that with appropriate caching this only requires re-running the mask-decoders twice for each frame, which is less than 10% extra overhead. On all RefVOS datasets we report J&F scores (Perazzi et al., 2016) (mean of IoU and contour accuracy) averaged on all text queries. Results on RefVOS-YTVOS and MeViS were obtained using the official test servers.

**Results.** Results are presented in Tab. 4. Our vanilla VoCap model (without FindTrack) outperforms the state-of-the-art on the challenging MeViS dataset by 2.4%, on RefVOS-DAVIS by +0.9%, and on UVO-VLN by 9.4%. On RefVOS-YTVOS, GLEE (Wu et al., 2024b) is slightly better by 0.3%. Now GLEE does tracking by detection, which requires making predictions for all frames before it runs an algorithm to merge these per-frame predictions into masklets; it is an offline algorithm which needs to analyze the whole video first. In contrast, our model runs in streaming fashion which makes it

Table 5: **Effectiveness of the SAV-Caption training data.** We show results for both video object captioning (mask-to-text) and referring object segmentation (text-to-masklet). Our SAV-Caption training set improves both tasks.

| | SAV-Caption-val (manual) | | RefVOS-YTVOS |
|---|---|---|---|
| | Captioning | SS-VOS | RefVOS |
| | CIDEr | J&F | J&F |
| Full training (Sec. B) | 47.8 | 75.5 | 70.3 |
| ↪ using 50% captions of SAV-Caption-train | 42.1 | 75.3 | 70.0 |
| ↪ using 0% captions of SAV-Caption-train | 27.4 | 75.6 | 66.6 |

more applicable in practice but this is a harder task. To overcome this, we can use FindTrack (Cho et al., 2025) at inference time which also turns VoCap into an offline model. This yields significant boosts on RefVOS-YTVOS (+0.9%) and MeViS (+1.1%) and sets a new state-of-the-art on these datasets. Hence we conclude that we outperform the state-of-the-art on RefVOS for all datasets.

We believe there are two main reasons why our model outperforms the state-of-the-art. First of all, we will demonstrate in Sec. 5.3 that we obtain synergies by sharing the text module between captioning and referring expressions. This means that the referring expression capabilities benefit from training captioning on the huge number of captions we provide through our SAV-Caption train dataset. Second, we are the only end-to-end trained model that performs referring expression segmentation through temporal propagation: DsHmp (He & Ding, 2024), UniRef++ (Wu et al., 2023), SOC (Luo et al., 2024), and GLEE Wu et al. (2024b) perform tracking by detection in which per-frame predictions are stitched together to form a masklet. But the memory-based, temporal propagation method of SAM2 (Ravi et al., 2024) was shown to outperform the SOTA tracking by detection method DEVA (Cheng et al., 2023a) on semi-supervised visual object detection, demonstrating the strength of such approach. FindTrack (Cho et al., 2025) and SAMWISE (Cuttano et al., 2025) are temporal propagation methods based on SAM2 (Ravi et al., 2024) but are not trained end-to-end.

### 5.3 Ablation on the effectiveness of SAV-Caption-train

To better understand the importance of our automatically annotated dataset (Sec. 3), we ablate its effectiveness. In particular, we compare our full training scheme with results where we use 50% and 0% of the captions in SAV-Caption-train, where not using any captions simply means we train on the original SAV dataset (Ravi et al., 2024). However, since SAV-Caption is our only video object captioning source, when not using any SAV-Caption data we instead invert RefVOS to become a captioning dataset (following (Zhou et al., 2023)): we consider the query text prompt, which is normally an input, as the output caption for the object. Tab. 5 shows the results.

On the SAV-Caption-val set, the CIDEr score goes down significantly when using only half the captions, and almost completely collapses without any SAV-Caption-train captions. This demonstrates that our automatic annotation is essential to obtaining good captioning performance on SAV-Caption val. Instead, as expected the lack of captions hardly impacts the SAV SS-VOS performance as this task does not use language at either the inputs or outputs.

Interestingly, on the RefVOS-YTVOS dataset we observe a significant increase from 66.6% to 70.3% J&F when training on the captioning task using SAV-Caption-train. This validates our design choice to share the weights of our language module for both the input text prompts and the captioning task: our model is able to exploit the synergies between referring expression segmentation and object captioning, and allows the referring expression task to benefit from training captioning on the huge amount of captions which we generated Sec. 3.

## 6 Conclusion

We proposed a video object segmentation and captioning model that takes either a box, mask or text prompt as input. We manually collected evaluation data for this task, and proposed an automatic annotation pipeline to curate training data. VoCap trained on our SAV-Caption dataset together with diverse existing datasets establishes a benchmark for video object captioning and outperforms the state-of-the-art on referring expression video object segmentation. We hope our model and datasets provide a foundation for fine-grained spatio-temporal video understanding, and encourages more work in this direction.

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

# A    DETAILS ON DATASET

## A.1    QUALITY OF SAV-CAPTION-TRAIN

We performed a quantitative evaluation on the quality of the SAV-Caption-train set by having the authors examine captions of 50 randomly selected objects from 50 different videos. They verified separately whether each of the elements used from our structured prompt was correct or not: the object category, its properties, and what the object does (e.g. motion or action). Furthermore, we counted how many properties and actions were obvious yet not generated in the caption. Results are in Tab. 6.

Table 6: Quantitative analysis of the quality of SAV-Caption-train on 50 objects in 50 different videos. # evaluated means the number of respectively categories, properties, and motion/actions we evaluated.

|  | correct | incorrect (hallucinations) | # evaluated | # missing aspects |
|---|---|---|---|---|
| object category | 88.0% | 12.0% | 50 | - |
| object properties | 87.6% | 12.4% | 105 | 7 |
| object motion / action | 85.5% | 15.5% | 62 | 5 |

The object category was correct in 88.0% of the cases. When an object was incorrect it was either subtle (e.g. *sock* instead of *shoe*) or it was a piece of clothing worn by a human and the human was captioned instead. Properties are also correct in 87.6% of the cases. Many mistakes were subtle color differences due to lighting conditions. When the human was described instead of their clothing worn, we counted these properties as incorrect (even if they were correct for the human). There were a few properties noticeably absent, mostly because of the context of other mentioned properties. For example, one caption mentioning a white striped sweater, whereas the sweater was *blue*-white striped, which conveys a quite different appearance of the sweater. The object's motion/action was correct in 85.5% of the cases. Most mistakes were subtle differences between standing still or driving / walking slowly. Similarly as before, when the motion/action was of the wrong category (person instead of sweater), we counted this as incorrect. In 7 instances we found an action to be clearly missing. This was usually when the object did multiple things sequentially (e.g. a person is first standing, then walking away) where one of them was missing. In another there was a parrot which was correctly identified to be laying on the floor, but they did this to scratch their head on the floor; a crucial aspect to understand its behavior.

To conclude, while there is some noise in the automatically generated data we consider it to be of decent quality. Moreover, in our main paper we clearly demonstrate the usefulness of this data for training captioning models.

## A.2    TEXT PROMPT TO GENERATE SAV-CAPTION-TRAIN

We use the following prompt together with our vision prompts to generate the pseudo-labels of our training set.

```
Describe the subject in the red contour in the following video.  If
the subject is a part of an object, please describe this part instead
of the whole object.  Please DO NOT DESCRIBE anything in the blurred
background outside the red contour.  First determine the subject's
category (CATEGORY), properties (PROPERTIES), action (ACTION), and then
give a description in ONE sentence (DESCRIPTION) including category,
properties, and action, etc..  Please use this FORMAT: 'The video shows a
CATEGORY. The subject's properties are PROPERTIES. The subject's action
is ACTION. DESCRIPTION.'.  The DESCRIPTION starts with 'A/ An CATEGORY'
or 'A/ An PROPERTIES CATEGORY' if it is grammarly more proper to put the
properties before the category.  The category, properties, motion and
the descriptions should be consistent.  PROPERTIES should be about the
objects appearance (color, texture, size, material, shape), what it is
wearing or a functional property (e.g.  fast, sharp).  Please always
include interesting or unexpected properties.  If there are multiple
```

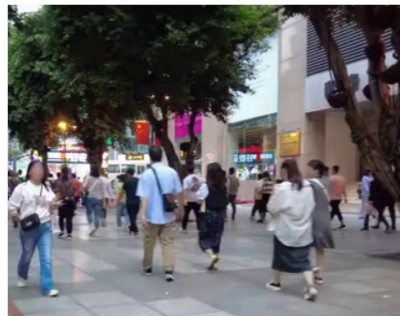

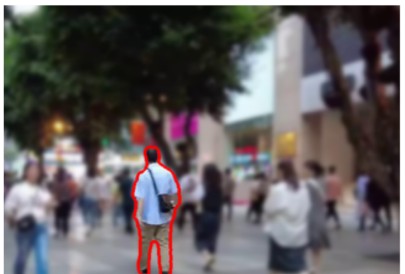

Figure 3: **Illustration of our visual prompting.** Top: the original frame; Bottom: our processed input to the Gemini annotator. We apply a red contour to highlight the target object and blur the background avoid distractions.

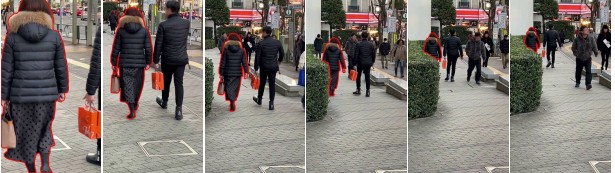

A woman wearing a black puffer coat with fur trim on the hood and a polka dot dress is walking

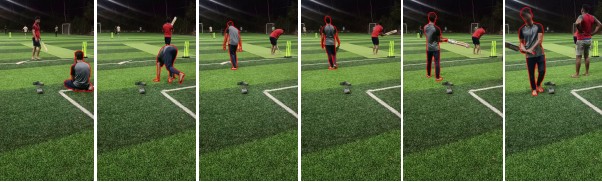

A barefoot person wearing a gray t-shirt and dark pants gets up from kneeling, picks up a cricket bat, and then stands

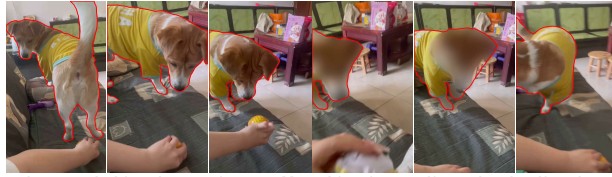

A brown and white dog wearing a yellow shirt is standing, then bending down to smell an orange, then standing again

Figure 4: **Examples of our VLM pseudo-labeled SAV-Caption training set.** We show the target object in red contour, and show the output captions below. The captions capture object classes, appearance properties, and multiple actions connected by "then".

```
actions happening sequentially, connect them with 'then', but do not
include more than 3 actions.  For static objects or parts, just say the
ACTION is 'static' and it is OK to not include ACTION in DESCRIPTION.
Please DO NOT mention the red contour in the description.  If the subject
is a person, please avoid describing the person's skin color and describe
the person's clothes color instead.  You only need to describe the
details that you are certain about.  If you cannot perform the task or
you are very uncertain, please say 'I cannot perform the task for this
video.'.
```

# B  IMPLEMENTATION DETAILS

Our image encoder is EVA02-L Fang et al. (2023), a 24-layer ViT model Dosovitskiy (2021) with MAE He et al. (2022) and CLIP Radford et al. (2021) pretraining. We chose this encoder as it is more suitable for language tasks, compared to the MAE-pretrained ViT used in SAM (Kirillov et al., 2023; Ravi et al., 2024). Our shared language encoder and decoder is a 6-layer BERT model (Devlin et al., 2019) with random initialization, shown to be effective and efficient for object captioning (e.g. Wu et al. (2024a); Zhou et al. (2023); Wang et al. (2022)). The text feature extractor contains 2 cross-attention layers with the same architecture as the mask decoder. Other modules follow SAM2 (Ravi et al., 2024) and are randomly initialized. Appendix C shows that our re-implementation of SAM2 is comparable to the original, and that EVA02-L is a strong alternative backbone.

Since we use an EVA02-L backbone, we cannot use existing SAM2 checkpoints. Instead, we pre-train our visual components on SAV Ravi et al. (2024), YTVOS Xu et al. (2018), and DAVIS Perazzi et al. (2016), following the SAM2 data mixture ratio (49.5: 9.2: 1.3). We train for 300k iterations, using a batch size 64 at $512 \times 512$ resolution. We verified that this training recipe produces results close to the official SAM2 model which is trained on proprietary datasets and uses a larger resolution (see Appendix C for more details). For the text encoder and decoder we use existing model weights trained for image captioning on WebLI Chen et al. (2022). After pre-training we train jointly on SAV-Caption-train (captioning and SS-VOS task), RefCOCO Yu et al. (2016), RefVOS-YTVOS Seo et al. (2020), and Visual Genome Krishna et al. (2017b) (captioning task), with a mixture ratio of 4:2:2:1, for 240k iterations with batch size 32. We finish with a small fine-tuning stage per dataset.

## C SAM2 BASELINE DETAILS

The original SAM2 (Ravi et al., 2024) was trained on private datasets in addition to the publicly-released SAV training and validation set. The publicly-released SAM2 training code[1] includes finetuning pipeline on MOSE dataset (Ding et al., 2023b), but does not include the main training loop. Therefore, before adapting SAM2 in our use case, we attempt to reproduce SAM2 training in our framework in Jax (Bradbury et al., 2018; Dehghani et al., 2022). We also repleace the MAE-pretrained backbone Hiera (Ryali et al., 2023) with a more vision-language native backbone Eva02 (Fang et al., 2023). When using Eva02, we reduce the input resolution from the original 1024 to 512 to fit our hardware, and verified minimal performance drop compare to the official SAM2 with Hiera-T and 1024 input size. We do not use the SA1B dataset (Kirillov et al., 2023) for pretraining as we did not find it helpful in our target datasets. We adapt the training hyper-parameters in Table 12 (b) of the SAM2 paper, which we summarize in Tab. 8.

Table 7: **Results of our reproduced SAM2**. We use a vision-language native backbone Eva02 (Fang et al., 2023) with a smaller input size (512 vs. 1024), and show the performance matches the original SAM2.

|  | backbone | resolution | MOSE-dev | SAV-val |
|---|---|---|---|---|
| Official SAM2 | Hiera-L | 1024 | 77.9 | 77.9 |
| Official SAM2 | Hiera-T | 512 | 75.3 | 75.2 |
| Our reproduction | Hiera-T | 512 | 76.9 | 74.8 |
| Our reproduction | EVA02-L | 512 | 75.7 | 75.8 |

As a result, our reproduced SAM2 with Eva02 (Fang et al., 2023) and a smaller input size trained on public data closely matches the official released model, as shown Tab. 7.

## D TRAINING HYPER-PARAMETERS

We include the full hyper-parameters used during training in Tab. 9

---

[1]https://github.com/facebookresearch/sam2

Table 8: **Hyperparameters of our reproduce of SAM2 as our pre-training.** We follow SAM2 (Ravi et al., 2024) for most hyperparameters.

| config | value |
|---|---|
| data | SA-V, YTVOS, DAVIS |
| data-ratio | 49.5: 9.4: 1.3 |
| steps | 300k |
| backbone | Hiera-T / Eva02 |
| resolution | 896 (Hiera-T) / 512 (Eva02) |
| optimizer | AdamW |
| optimizer momentum | $\beta_1, \beta_2$=0.9, 0.999 |
| gradient clipping | type: $\ell_2$, max: 0.1 |
| weight decay | 0.05 |
| learning rate (lr) | img. enc.: 4e-5, other: 4.0e-4 |
| lr schedule | cosine |
| warmup | linear, 1k iters |
| layer-wise decay | 0.8 |
| augmentation | hflip, crop and square resize to 512 |
| batch size | 64 |
| drop path | 0.1 (Hiera-T) / 0.4 (Eva02) |
| mask losses (weight) | focal (20), dice (1) |
| IoU loss (weight) | $\ell_1$ (1) |
| occlusion loss (weight) | cross-entropy (1) |
| num frames | 8 |
| max. masks per frame. | 2 |

| config | value |
|---|---|
| data | SAV-Caption, RefVOS-YTVOS, RefCOCO, VisualGenome |
| data-ratio | 2: 1: 1: 0.5 |
| steps | 240k |
| backbone | Eva02 |
| resolution | 512 |
| optimizer | AdamW |
| optimizer momentum | $\beta_1, \beta_2$=0.9, 0.999 |
| gradient clipping | type: $\ell_2$, max: 0.1 |
| weight decay | 0.05 |
| learning rate (lr) | 5e-5 |
| lr schedule | cosine |
| warmup | linear, 1k iters |
| layer-wise decay | 0.8 |
| augmentation | crop and square resize to 512 |
| batch size | 32 |
| drop path | 0.4 |
| mask losses (weight) | focal (20), dice (1) |
| IoU loss (weight) | $\ell_1$ (1) |
| occlusion loss (weight) | cross-entropy (1) |
| caption loss (weight) | cross-entropy (1) |
| caption loss label smooth | 0.1 |
| num frames | 8 |
| max. masks per frame. | image: 32, video: 2 |

Table 9: **Hyperparameters of next stage of VoCap training.** We continue train on datasets with both text and mask annotations on both images and videos, with a cross-entropy caption-loss.

