# Rater instructions
## For object-centric video captioning (on the SAV dataset)

# The task: Describe the object outlined in red

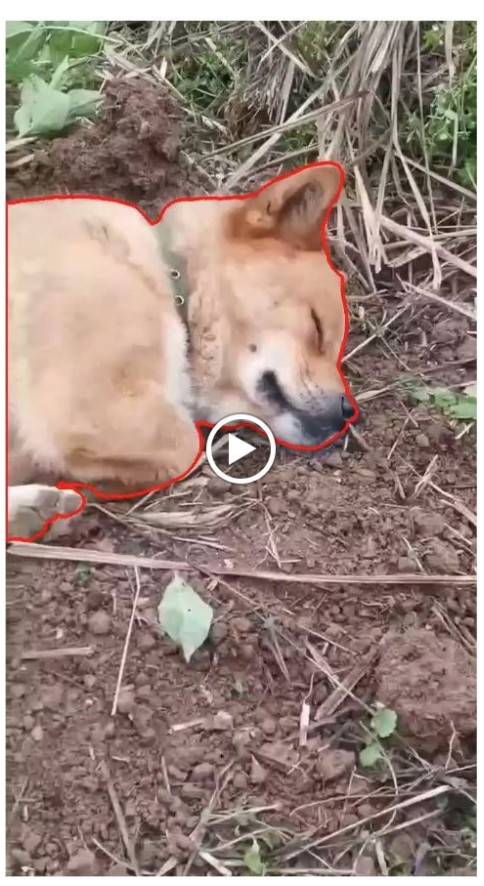

Watch the video. Then create a caption for the object in the video which is outlined red. For example:
1. The brown dog is laying on the ground.
2. The dog, which is brown and has a green collar, is laying on the ground.

We want captions to describe the object which is outlined. Ideally, the caption should include:
1. The object which is outlined.
2. Properties of the object, such as its appearance, what it is wearing or a functional property (e.g. fast, sharp). *Always include interesting or unexpected properties.*
3. What happens with the object. For example, the object can act itself: (e.g. the dog lays down), or may be acted upon (e.g. the bread is eaten).

# Take care when describing persons

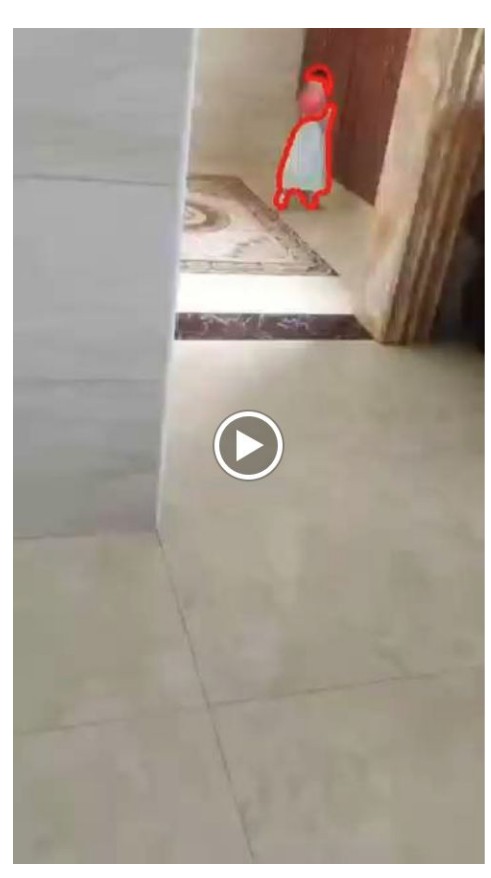

Use gender and age neutral words to describe humans.

*Good caption:*
- A person, who is wearing a light-blue dress, walks with a ball and then drops it on the floor

*Bad caption:*
- A girl [...]
  - Be gender neutral and age neutral.
- A chinese-looking child [...]
  - Don't refer to any racial features. Hence don't use terms like 'white', 'dark', etc.
    Note: if you want to refer to the clothing, be explicit:
    - Don't say: "a white girl".
    - Say: "a girl, who wears a white dress,"

# Capture interesting properties

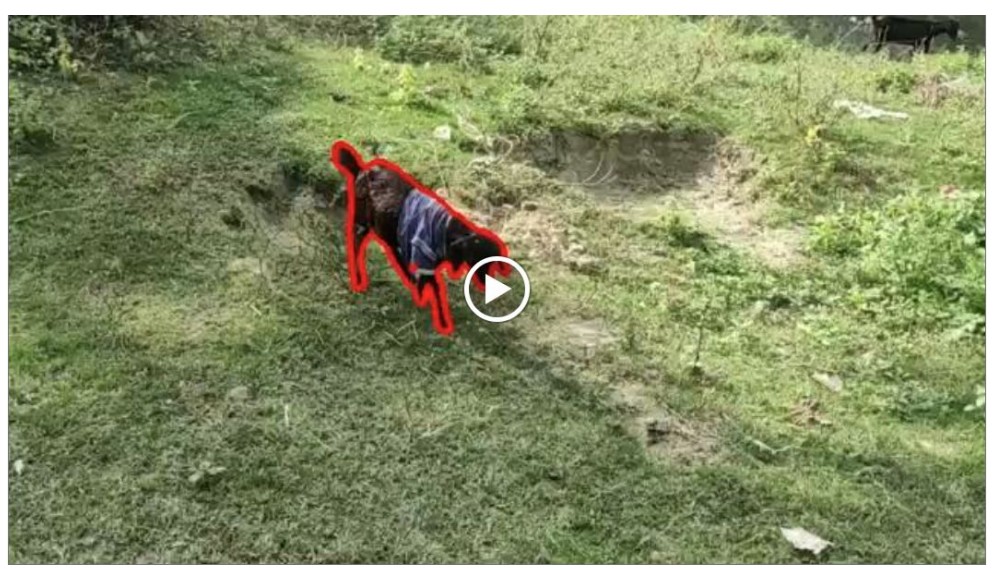

*Good caption:*
- A sheep, which is brown and wears a blue t-shirt, walks over the grass towards another sheep and eats a bit on the way.
- A sheep with a blue blue shirt walks over the grass.

*Bad caption:*
- A brown sheep walks [...]
  - The t-shirt stands out here and should be captured.

# Keep captions to a single sentence

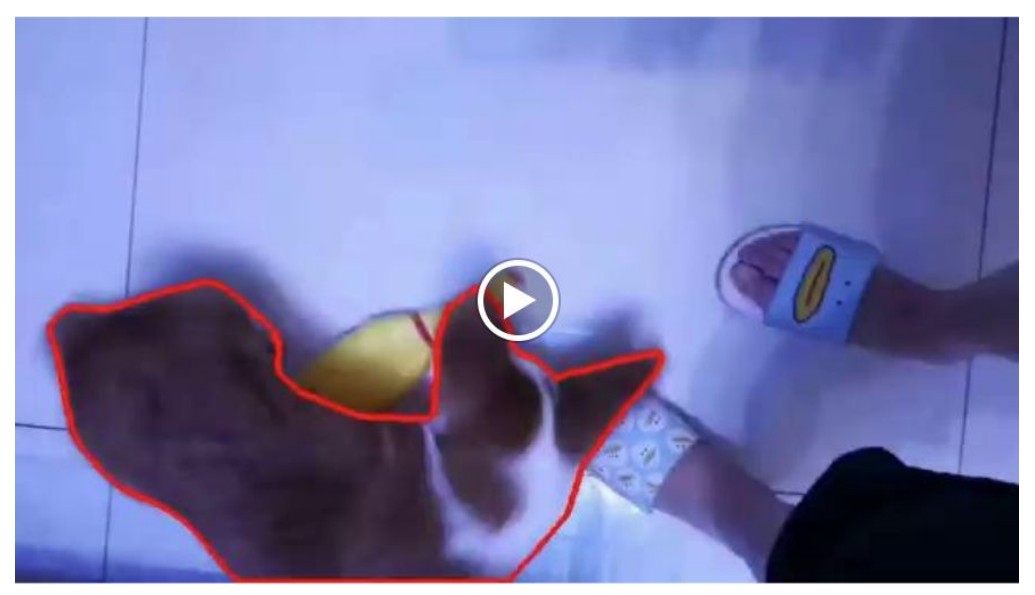

*Good caption:*
- A white-brown dog plays with a rubber chicken and then with a ball.
- A white-brown dog plays first with a rubber chicken and afterwards with a ball.

*Bad caption:*
- A white-brown dog plays with a rubber chicken. Next, it starts playing with a ball.
- There is a dog which is brown with a bit of white. It first plays with a rubber chicken. Then it plays with a ball.

# Don't describe irrelevant general properties of the scene

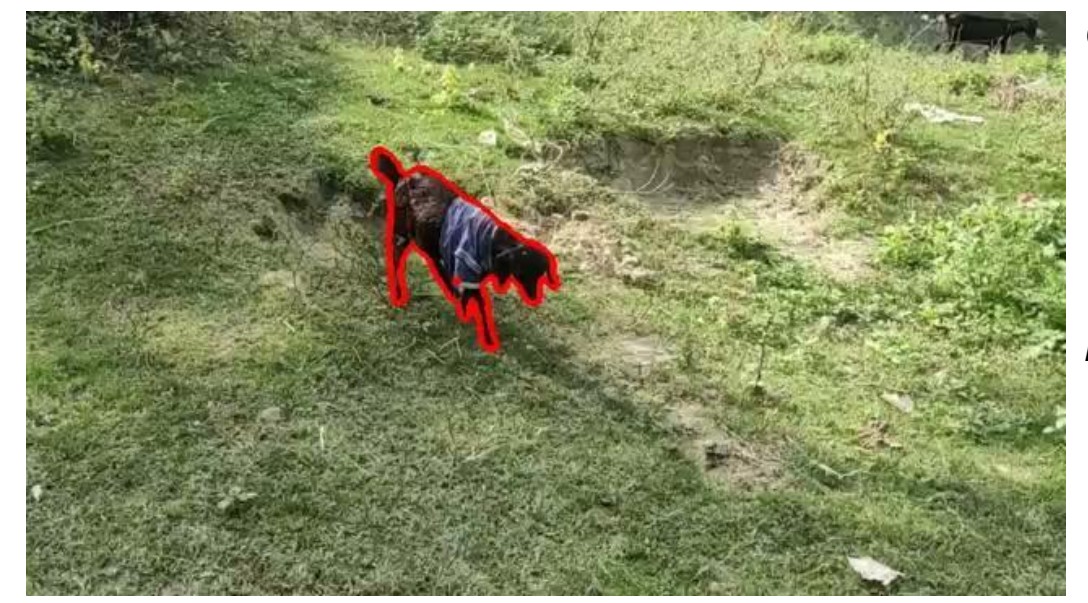

*Good caption:*
- A sheep, which is brown and wears a blue t-shirt, walks over the grass towards another sheep and eats a bit on the way.
- A sheep, which is wearing a blue shirt, walks over the grass.

*Bad caption:*
- A sheep, which wears a blue t-shirt, walks over the grass and in the background there is a pond with birds in it.
  - Don't describe the background if it is irrelevant to the object.
  - However, *do* describe objects if they are relevant. For example, if you feel that the sheep is intentionally walking towards the pond, you could say: "the sheep walks to the pond".

# Describe only things relevant to the object itself

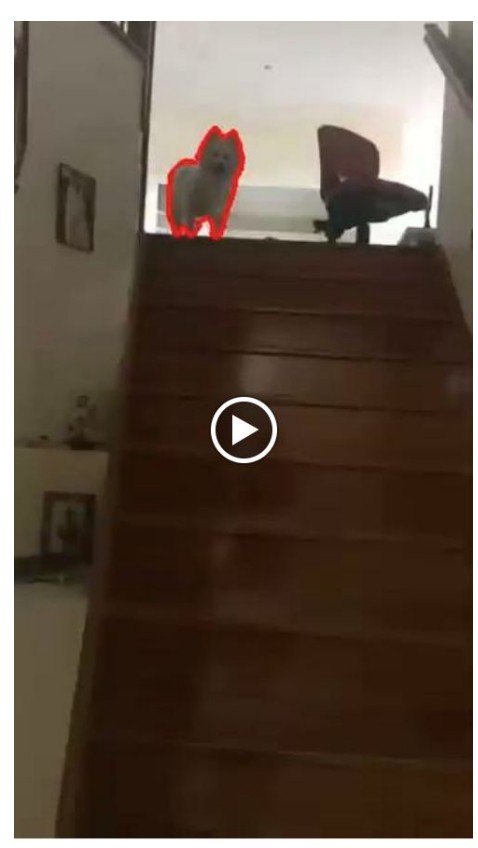

*Good caption:*
The white dog runs down the stairs and then jumps in a car.

*Bad caption:*
The white dog stands next to an office chair which is not moving and then runs down the stairs where paintings are hanging on the walls then passes a hall with a shoe rack, turns into a room which is probably the garage and then jumps in a car.
- The office chair seems irrelevant, as does its movement.
- The paintings on the wall don't have any relation to the dog.
- The shoe rack is an unnecessary detail since the dog does not interact with it.

# Only describe properties related to the object

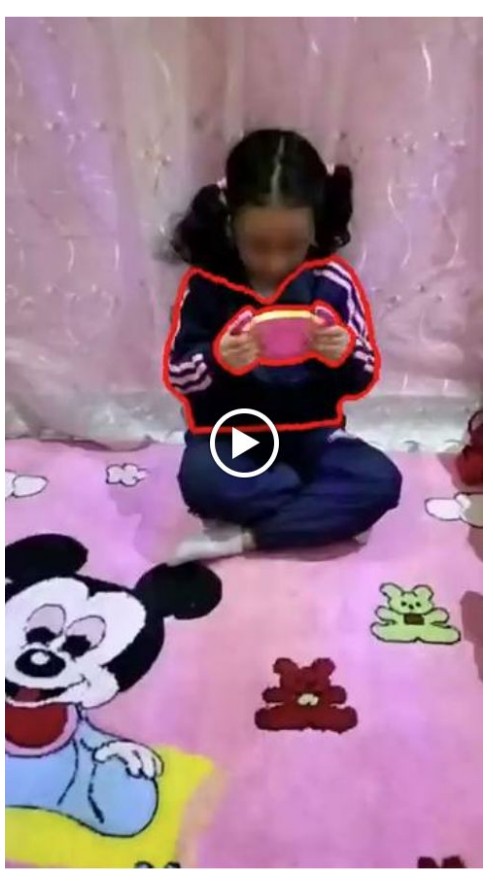

*Good caption:*
- A blue sweater with white stripes on a person.
- A sweater, which is blue with white stripes, is worn by a person.

*Bad caption:*
- A sweater, which is blue with white stripes, is worn by a person which is playing a video game on a handheld.
  - Don't describe what the person who is wearing the sweater is doing.
  - Unless what the person is doing has direct impact on the movement of the object. For example: if the outlined object is a hand, 'picking up' should be mentioned.

# Keep it simple for static objects

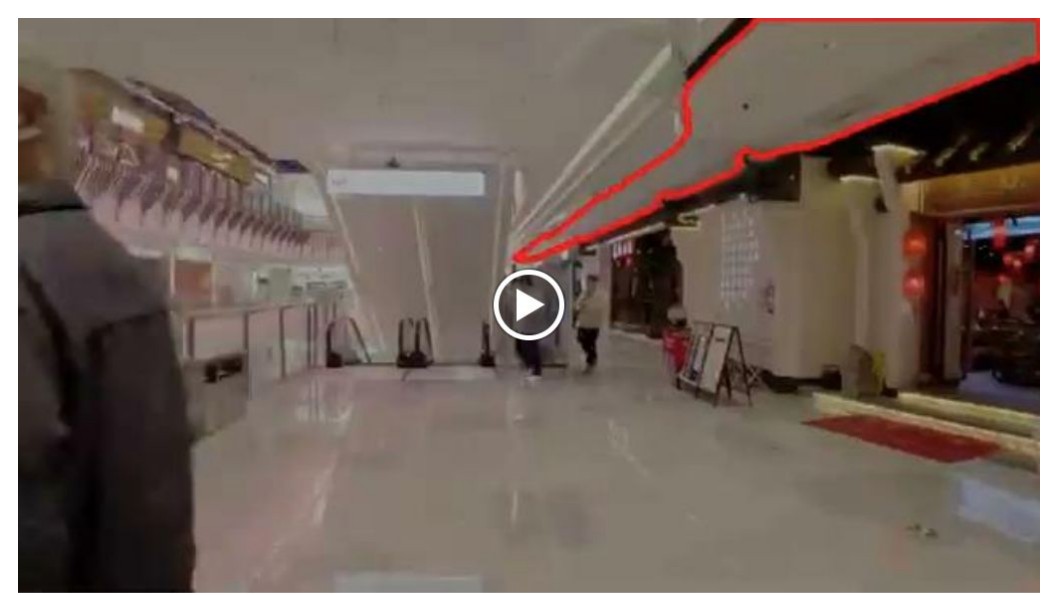

*Good caption:*
- A white ceiling of a shopping mall.

*Bad caption:*
- A white ceiling hangs in a shopping mall above pedestrians.
  - The ceiling does not interact with the pedastrians. This makes for an unnatural caption.

# Sometimes the object requires a longer description

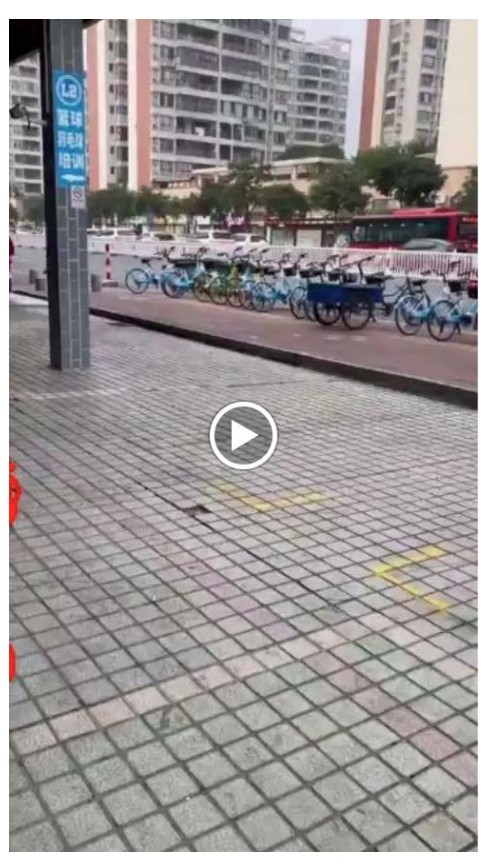

*Good caption:*

A person on a scooter, wearing a white helmet, drives towards the bicycle lane.

- While 'a person on a scooter' is not a single object, both the person and the scooter are outlined and there is no single word for this object.

*Bad caption:*

- A person, with a white helmet, drives towards the bicycle lane on a scooter.
  - Please keep the description of what is outlined in one place in the sentence.
  - This would be a correct caption if only the person was outlined.

*Discouraged caption:*

- A person, with a white helmet, riding a scooter drives towards the bicycle lane.
  - If possible, try not interleave the properties of the object with the description of the object.

# Appendix: Types of properties

- **Physical properties**; These are observable and measurable characteristics
  - Color
  - Texture
  - Size
  - Material
  - Appearance beyond color such as shininess
  - Shape
- **Functional properties;** which relate to what an object does or how it is used.
  - Expressed by adjectives such as sharp, fast, etc.
  - Expressed by verbs such as walking, running, turning left, drinking, etc.
    - A person, who is *holding* a bottle, is dancing in the karaoke bar.