# OpenReview forum: "VoCap: video object captioning and segmentation from any prompt"
_ICLR.cc/2026/Conference — Submitted to ICLR 2026_

### Official Review · Reviewer_18P3 · 2025-10-29

**Soundness:** 2
**Presentation:** 2
**Contribution:** 3
**Rating:** 4
**Confidence:** 4

**Summary:**

This paper proposes VoCap, a flexible video model that can receive video and multimodal prompts, output spatiotemporal masks and target guidance, and simultaneously solve promptable video target segmentation, referential expression segmentation, and target captioning tasks. To address data scarcity, the SAV-Caption dataset is constructed from the existing large-scale segmentation dataset SAV, and the model is trained at scale using additional image/video datasets. Experiments show that VoCap establishes a benchmark in video target captioning and image localization captioning.

**Strengths:**

1. VoCap breaks through the limitations of existing video understanding models that are "single input/single output" and achieves deep unification of multimodal inputs and outputs. It is the first model that can simultaneously cover three key tasks (video object segmentation, representation expression segmentation, and object captioning).
2. To address the industry challenge of "high cost and small scale of video mask + caption annotation," the paper constructs the SAV-Caption dataset, which surpasses existing datasets in scale, quality, and correlation strength, providing crucial data support for video object understanding tasks.

**Weaknesses:**

1. The SAV-Caption training set relies on pseudo-labels generated by Gemini 1.5 Pro. While visual cues (red outlines, blurred backgrounds) improve quality, these pseudo-labels introduce significant noise that may affect the model's generalization.
2. SAV-Caption is built on the SAV dataset, but SAV's scene coverage does not explicitly mention "extreme environments" (e.g., low light, heavy rain) or "fine-grained categories" (e.g., different animal species, complex mechanical parts). Compared with existing datasets (such as MeVIS with motion expression scenes), SAV-Caption's scene complexity remains limited, which may lead to lower model performance in unconventional scenes.
3. During model inference, the final caption is taken from the last frame of the video. Although the features of historical frames are fused through the memory module, key dynamic information of intermediate frames may still be lost. For example, when the target has multiple stages of action, such as "standing → walking → jumping", relying solely on the output of the last frame may simplify or omit the preceding actions.

**Questions:**

See Weaknesses

---

> ### Author Response · Authors · 2025-11-26
>
> We thank the reviewer for their comments. First of all, we would like to highlight that our model is state-of-the-art in referring expression segmentation, as is evidenced in Table 4. We feel this is a significant achievement which should be taken into count for rating our paper. We address the other points below.
>
> **Pseudo labels**
>
> We acknowledge that pseudo-labels can be noisy. In fact, we quantified this in Table 6 of the appendix where we show that 88% of object categories are correct. But training from noisy data is increasingly common in our field (see the pseudo labels section of our related work Sec. 2). And while better data will result in better models, we do not see how this would affect generalization. More importantly, the SAM2+Gemini Pseudo labeling results in Tab 2 demonstrate that our model can overcome part of the label noise in our data, as also discussed in L365-377.
>
> Interestingly, given that our data annotation pipeline is based on sending video data to a foundation model, the pseudo labels can be improved by upgrading the model. In fact, we did this after our original submission and indeed our captioning results improved significantly:
>
> **Table:** *CIDEr scores on SAV-Caption-val*
>
> | Method | Gemini 1.5 pseudo-labelling (Table 2) | Gemini 2.5 pseudo-labelling | Gain from updated Gemini |
> | :--- | :---: | :---: | :---: |
> | UniRef++ & Gemini | 34.3 | 40.7 | +6.4 |
> | SAM2 & Gemini | 40.5 | 63.2 | +22.7 |
> | VoCap trained on Gemini pseudo-captions | 47.8 | 72.2 | +24.4 |
>
> As foundation models become more powerful, we expect more papers to call these models in various ways to pseudo labels of increasing quality. We see this as an important direction in our field.
>
> **Complexity of the SAV dataset**
>
> The SAV dataset is by far the largest video segmentation dataset available, and therefore models trained on it may generalise the best. SAV does indeed not explicitly cover extreme environments or unconventional scenes, but due to its size it should include some of them. For example, low-light environments are relatively frequent. Due to its sheer size, we feel annotating SAV was the most useful for the community and the first thing to do. Furthermore, we point out that our pseudo-labelling pipeline can be applied to any video dataset, and is not specific to SAV. Therefore, we can easily extend it to a more suitable dataset if one becomes available.
>
> **Caption for the video is computed from the last frame**
>
> The reviewer is correct in that the caption is predicted from only the final frame, and information may have been lost by this point. However, since our model is trained end-to-end, it learns to extract and retain the relevant information for captioning at the last frame. The reviewer is correct that using representations from earlier frames may help, especially for long-running actions. However we believe that this approach is not as elegant or generalisable as simply using the last frame and defer further investigation to future work.

---

### Official Review · Reviewer_p7Gk · 2025-10-30

**Soundness:** 4
**Presentation:** 3
**Contribution:** 4
**Rating:** 8
**Confidence:** 4

**Summary:**

This paper presents VoCap, a strong general model that, given a video and an arbitrary prompt (text/box/mask), simultaneously outputs the target’s spatio-temporal segmentation (masklet) and an object-centric caption. To support training and evaluation, the authors build SAV-Caption, a data pipeline that generates video object-centric captions using a VLM from SAV, with an additional human-annotated validation set. Across both image and video settings, VoCap achieves or surpasses state-of-the-art results on object captioning and referring-expression video object segmentation.

**Strengths:**

1. The paper unifies prompting→segmentation and prompting→captioning within one framework, covering RefVOS, promptable segmentation, and location-conditioned captioning. Experiments show that training the location-conditioned captioning task benefits video object segmentation.
2. The SAV-Caption training set contains ~50k videos and ~170k objects (avg. 11.8 words per caption). Pseudo-labels are mask-grounded rather than box/track-grounded, offering a solid basis for future video segmentation + captioning research.
3. The method leads on RefVOS-DAVIS/MeViS/UVO-VLN, and reaches new SOTA on YTVOS when combined with FindTrack. On Visual Genome (box-conditioned captioning), it also outperforms recent methods.
4. The paper specifies a two-stage training schedule (SAM2 reproduction → multi-task joint training), dataset mixing ratios, and full hyperparameters, and states an intention to release the dataset. Releasing training code and data-processing scripts would further strengthen the impact.

**Weaknesses:**

No major flaws. My concerns and suggestions are listed below under Questions.

**Questions:**

The training data quality heavily relies on Gemini-generated captions. Although the pipeline uses constrained prompts plus two visual prompting techniques and includes small-scale human checks, systematic biases (e.g., small-object omissions, “actor bias”) may affect style and value alignment. Suggestions:
Add a more systematic human audit with error-type distributions, and compare/aggregate pseudo-labels from multiple VLMs.
Consider using a small set of human-labeled in-context examples to better constrain Gemini’s generations.
Beyond reporting average caption length, analyze linguistic diversity (e.g., lemma distributions, voice—active/passive, temporal connectives) and compare with the human validation set.

The architecture is SAM2-like and supports online inference, but the paper lacks throughput/latency/memory comparisons. FindTrack converts inference to an offline mode and achieves SOTA; the paper notes FindTrack adds <10% overhead, but please provide end-to-end throughput, latency, memory, and power metrics—especially on long videos.

The current model and dataset are primarily single-object. Please report failure modes and possible multi-object extensions. For example, when one description refers to multiple instances, what captions/masks are produced? Could multiple object tokens or coordinated prompts help?

Only CIDEr is used for captioning. Since video object captioning stresses attributes and temporal dynamics, please add SPICE / RefCLIPScore / GPT-judge or human evaluations (consistency, readability, temporal coverage). Also, VoCap’s video caption is taken from the last frame, whereas pipeline baselines vote across per-frame captions; please include a last-frame vs. voting ablation to align evaluation protocols and rule out methodological bias.

---

> ### Author Response · Authors · 2025-11-26
> **Please find our answers below**
>
> Thank you for your positive review and excellent suggestions.
>
> **Better generation of training data**
>
> These are all great suggestions. After our submission we went through the process of re-annotating our data with Gemini 2.5. To adhere better to the human styles we optimized our prompt and experimented with different visualization styles on a small separate human annotated held-out set of 185 captions. More specifically we did this by measuring caption alignment in terms of CIDEr, Rouge-L, and Bleu-4. We indeed found that adding some human-labeled in-context examples improved results. In addition, we did some more systematic ablations on how to feed the data to Gemini: we used masks, boxes (which is also typical computer vision data), and circles (which seems more natural for a human). We found that masks worked best. In addition and in contrast to what we did in our submission, blurring the background actually reduced alignment (at least for Gemini 2.5 pro; Gemini 1.5 pro is not available anymore via the public API). The ablation is presented below.
>
> | Visualization type | CIDEr | Rouge-L | Bleu-4 |
> | :--- | :---: | :---: | :---: |
> | Red boxes | 54.3 | 39.3 | 13.6 |
> | Red circles | 61.1 | 41.2 | 14.5 |
> | Red masks | 66.9 | 45.3 | 17.1 |
> | Red masks, blur background | 59.6 | 43.3 | 16.4 |
>
> Using red masks without blurring the background, we re-annotated the full SAV-training set. Then we re-trained our VoCap model and re-did the strong baseline experiments of Table 2. While J&F stayed mostly the same for VoCap (it slightly increased to 75.7), captioning results improved significantly.
>
> **Table:** *CIDEr scores on SAV-Caption-val*
>
> | Method | Gemini 1.5 pseudo-labelling (Table 2) | Gemini 2.5 pseudo-labelling | Gain from updated Gemini |
> | :--- | :---: | :---: | :---: |
> | UniRef++ & Gemini | 34.3 | 40.7 | +6.4 |
> | SAM2 & Gemini | 40.5 | 63.2 | +22.7 |
> | VoCap trained on Gemini pseudo-captions | 47.8 | 72.2 | +24.4 |
>
> We will update our paper with these new Gemini 2.5 captions. We will also do a more thorough linguistic analysis of the human/vocap/gemini annotation pipeline captions.
>
> **Single-object model**
>
> The current model is indeed single-object only and trained exclusively for a single object. Hence if a query is ambiguous we saw that while it sometimes starts with multiple objects, the tracking behaviour is strong and the mask quickly converges to focus on a single object. For a multi-object extension, it would be possible to start from generic object tokens which look at the input prompt, similar to DETR (Carion, 2020).
>
> **Other captioning and latency metrics**
>
> Thank you for the suggestion, we will add these other captioning and latency metrics to the paper.
>
> **Aggregating captions**
>
> The baseline methods in Tab. 2 are actually given the whole annotated video to output a single caption. For VoCap, it is theoretically possible to output a caption at every single frame but it is unclear how to aggregate these into a single caption.

---

> > ### Author Response · Authors · 2025-12-03
> > **Latency metrics**
> >
> > We calculated latencies on a TPUv6e to more precisely measure the computational requirement for FindTrack inference. Findtrack requires (i) extracting image features (plus minor initialization overhead), (ii) creating mask predictions from all image features individually, where no memory is involved, and (iii) creating mask predictions in a tracking fashion, which requires memory-attended image features and updating the memory. In contrast, the normal VoCap inference mode requires (i) and (iii). Exact timing measurements for these phases are as follows:
> >
> > Extracting image features + minor additional overhead: 6.6 ms / frame
> > Individual frame predictions (no memory): 4.2 ms / frame
> > Tracking-based frame predictions (with memory): 9.6 ms / frame
> >
> > This results in 16.2 ms/frame for standard VoCap or 62 fps, and 20.4 ms/frame or 49 fps for VoCap with FindTrack. It does require holding all image features in memory for all frames, which is 3MB / frame. Now the total memory consumption of VoCap inference is roughly 2GB. So even for a 1000 frame video the footprint will be 5GB, which is well below the current standard for modern graphic cards. We will add this to our final version of the paper.

---

### Official Review · Reviewer_UGYr · 2025-10-31

**Soundness:** 3
**Presentation:** 3
**Contribution:** 3
**Rating:** 6
**Confidence:** 5

**Summary:**

Summary:
VoCap proposes a unified model for video object description and segmentation, which can output both spatio-temporal segmentation masks (masklets) and object-level textual descriptions based on any input prompt (text, box, mask). It achieves state-of-the-art performance on referring expression video object segmentation (VOS).

**Strengths:**

Advantages:
1. Efficient data utilization: By using pseudo-labels, the model significantly expands the training data and reduces manual labor costs.
2. The paper is clear and easy to follow.
3. Subtitle training improves the understanding of referring expressions, showcasing the advantages of language-vision collaboration.
4. Qualitative examples demonstrate the effectiveness of the method.

**Weaknesses:**

Disadvantages:
1. What is the key difference in terms of spatio-temporal reasoning (mask-level) between the proposed method and [1]?
[1] "VISA: Reasoning Video Object Segmentation via Large Language Models"
2. The argument that "there is yet no existing computer vision system that is capable of both spatio-temporal localization via segmentation masks, as well as a semantic understanding of objects via natural language" might be an over-claim.
3. The masklet (three separate temporal masks) could be presented in the prediction of Figure 2.

**Questions:**

1. In the VOS field, it is common to use the first-frame mask as the reference for mask-guided segmentation, while some other works use a point as the base. Which of these two methods is more natural in the VLM (Vision-Language Models) field? Can the framework proposed by the authors support a point as an additional input in the future?

2. It would be beneficial to add a "Broader Impact" section to discuss both the positive and negative impacts on the community. I recognize the importance of this field, but one question is that the paper already supports multiple modalities (text, box, mask) as inputs and dual outputs (mask + caption), offering considerable flexibility. Is this direction still the main focus of research in the field? Would further developing a more unified framework that supports more types of inputs have significant value? Additionally, is researching which types of user input could better adapt to existing technologies a potentially promising direction?

---

> ### Author Response · Authors · 2025-11-26
> **Please find our answers below**
>
> Thank you for the thoughtful review. Please find the answers to your questions below:
>
> **Q1: For VOS and VLMs, what should the first-frame reference be?**
>
> In the VOS field the most common input is a mask on the first frame. However, SAM2 introduced the possibility of using a point, which is common in the interactive segmentation literature in images. However, there are no proper evaluation datasets for video out there which support point-based inputs (as far as we know), nor did people repurpose existing VOS datasets where they reported baselines to compare to. That being said, it would be straightforward to train our model for input points since the location prompt encoder (L237) already supports it; only the training data needs to be adjusted.
>
> **Q2: Broader impact**
>
> We will add a broader impact section to the supplementary material. Generally there are quite a few papers which go in the direction of supporting multiple modalities. We cite some more image segmentation focused papers like GLEE (Wu et al., 2024b) and  Uniref++ (Wu et al., 2023), but there are also works which focus more on segmentation and VQA tasks such as the VISA work which you mentioned (thanks for the reference) and the Sa2VA paper mentioned by Reviewer Gtv6. It is still an open question how much synergy one can get from combining multiple tasks or if we should move more towards VLMs with tool calls to obtain a good general model.
>
> **W1: Comparison to VISA**
>
> VISA is a generalist referring expression segmentation framework which combines a frozen MM-LLM (LLaMA-VID) with a frozen visual encoder (SAM) and a frozen object tracker (XMem). The MM-LLM and the SAM parts are glued together using Lora finetuning and training the SAM mask decoder. The object tracker is fully separate. More specifically, given a video they select the ‘most distinguishing frame’ and decode the mask based on a token generated from an LLM. Then starting from this mask they track this using a pre-trained tracking model (XMem).
>
> In contrast, we train our full model end-to-end where the object tracker is fully integrated into the training procedure. Furthermore, the referring expression is used as context during the full tracking procedure. In our experiments with FindTrack we found this to be important: in preliminary experiments when using VoCap as pure VOS after the first frame reduced performance by 1-3%.
>
> Note that our results are significantly higher than VISA, which adds a mask decoder to a frozen LLM, whilst our model is trained fully end-to-end. We show these qualitatively in the table below, and will update our state-of-the-art comparisons in Table 4 accordingly.
>
> | Model | MeVis | Ref-YTVOS |
> | :--- | :---: | :---: |
> | VISA | 44.5 | 63.0 |
> | VoCap (ours) | 53.0 | 71.2 |
>
>
> **W2: Adjusting claim of “Computer vision system that is capable of both spatio-temporal localization via segmentation masks”**
> Fair point. We will rephrase and cite recent other contemporary work which does exactly this (e.g. Sa2VA).
>
> **W3: Masklet could be presented in Figure 2**
>
> Thanks for the suggestion. We will show this in the revision.

---

### Official Review · Reviewer_Gtv6 · 2025-10-31

**Soundness:** 2
**Presentation:** 2
**Contribution:** 2
**Rating:** 2
**Confidence:** 4

**Summary:**

In this paper, the authors propose a new dataset called SAV-Caption. Then they use the datasets to train the model VoCap. The authors evaluate both the models and datasets. However, based on the current results, I am not fully convinced with the performance of both the datasets and model.

**Strengths:**

The paper builds an auto-mated pipeline to label the data. It reduces the cost of human-labeling.

The authors evaluate the performance of both the model and the datasets.

**Weaknesses:**

1. The paper does not mention "Sa2VA" or the "Ref-SAV" dataset. Consequently, it lacks a direct comparison between its pseudo-labeling pipeline (which creates SAV-Caption) and the one used by Sa2VA. Given that the data labeling pipelines appear similar, the novelty of VoCap's contribution seems limited.

2. The paper omits "InstructSeg" and "Sa2VA" from its comparison tables. In Section 5.2 and Table 4, the authors compare VoCap's performance on Referring Video Object Segmentation (RefVOS) against models like UniRef++ and SAMWISE. Based on this limited comparison, the paper claims to "outperform the state-of-the-art on RefVOS for all datasets." However, the exclusion of known models like "InstructSeg" and "Sa2VA" makes this claim questionable.

3. To properly verify the effectiveness of the proposed datasets, the authors should have used more advanced models for co-training or finetuning. Currently, verification of the dataset's effectiveness is limited to the results in Table 5. Given that the model used for this verification does not achieve state-of-the-art performance, it is difficult to evaluate the true utility of the proposed datasets.

4. There are potentially alternative, and perhaps simpler, ways to label the SAV dataset. While the authors detail their chosen pipeline in Section 3 (using Gemini 1.5 Pro, Visual Prompting, and Background Obscuring), it would also be possible to use a model like the Describe Anything Model (DAM) to obtain object-level descriptions, which could form a dataset analogous to SAV-Caption.

Conclusion: Based on the current results, I am not fully convinced of the usefulness of the proposed datasets and model. Furthermore, the paper's state-of-the-art claim is questionable.



DAM: https://describe-anything.github.io/
InstructSeg: https://arxiv.org/abs/2412.14006
Sa2VA: https://github.com/bytedance/Sa2VA

**Questions:**

NA

---

> ### Author Response · Authors · 2025-11-26
> **Please find our response below**
>
> We thank the reviewer for their comments. First of all, while Sa2VA, InstructSeg and the Describe Anything Model (DAM) are highly relevant to our work, they are considered contemporary works since Sa2VA is only available on ArXiv and the latter two are published at ICCV’25 (October, after the ICLR deadline).
>
> **Comparison on the RefSAV dataset**
>
> Sa2VA created the RefSAV dataset on a part of the SAV training set using InternVL2-76B and Qwen2-72B. Since this is contemporary work, our data generation pipeline should be considered as novel as theirs as per [ICLR policy](https://iclr.cc/Conferences/2026/ReviewerGuide). Nevertheless, there are a few important differences. First of all, the RefSAV test set was created on top of the SAV-training set (see their Sec 3.3). This means one cannot fairly evaluate any model which included this data in its training, which excludes all models building on pre-trained SAM2 weights (!). In contrast, we annotated the SAV validation set manually, which makes it much more useful for the community. Second, Sa2VA pseudo-annotated only part of the SAV training set: 37k videos with 72k expressions. In contrast, we pseudo-annotated the full SAV-training set: 50k videos with 170k captions.
> To enable comparison of our model on their RefSAV dataset, we re-trained our model as before (details App. B) but swapped out the original SAV-caption training set for the RefSAV-train videos to avoid test-set contamination. In this setup we used their referring expressions as input prompts. We outperform Sa2BA-8B by 4.4 points J&F.
>
> | Method | J&F RefSAV short |
> | :--- | :---: |
> | Sa2VA-8b (from their Table 10) | 41.2 |
> | VoCap | 45.6 |
>
> **Comparison to InstructSeg and Sa2VA**
>
> We compare to the main results presented in Sa2VA (their Table 6) and InstructSeg (their Table 4) on MeVis and Ref-YTVOS below. We outperform both models substantially: by 6.1 points on MeVis and 0.5 points on Ref-YTVOS.
> In addition, both InstructSeg and Sa2VA combine an LLM with a vision encoder, resulting in models which are substantially larger than ours (3B-26B params). In contrast, VoCap  only has 416 million parameters. Since it has been trained specifically for video segmentation and captioning tasks, it outperforms models an order of magnitude larger in terms of parameters.
>
> | Model | Params | MeVis | Ref-YTVOS |
> | :--- | :---: | :---: | :---: |
> | Sa2VA-8B | 8B | 46.9 | 70.7 |
> | Sa2VA-26B | 26B | 46.2 | 70.1 |
> | InstructSeg | 3B | - | 67.5 |
> | VoCap | 416M | 53.0 | 71.2 |
>
> We will add these results to Table 4 of our paper. We also remind the reviewer that these are contemporary works which do not actually require comparison as per the [ICLR policy](https://iclr.cc/Conferences/2026/ReviewerGuide)
>
> **To verify the effectiveness of the proposed datasets, the authors should have used more advanced models**
>
> We demonstrated above an in our Table 4 that VoCap is state-of-the-art for referring expression segmentation. Hence this validates the ablation experiments in Table 5.
>
> **Other ways to obtain labeled data**
>
> The Describe Anything Model (DAM) was published at ICCV’25 (after the ICLR submission deadline) and is considered contemporary as per [ICLR policy](https://iclr.cc/Conferences/2026/ReviewerGuide). When we started our project there was no proper data available and therefore we generated this dataset which can be directly used by the community. In terms of utility, both DAM and our Gemini annotation pipeline require only videos with masks with otherwise trivial preprocessing. One advantage of our pipeline is that annotation quality immediately improves as foundation models improve, whereas DAM will need to be retrained. To illustrate this, we re-annotated SAV-train using Gemini 2.5 pro while also re-running our strong baselines (Tab.). While J&F stayed about the same (it slightly improved to 75.7), the captioning scores significantly increased.
>
> **Table:** *CIDEr scores on SAV-Caption-val*
>
> | Method | Gemini 1.5 pseudo-labelling (Table 2) | Gemini 2.5 pseudo-labelling | Gain from updated Gemini |
> | :--- | :---: | :---: | :---: |
> | UniRef++ & Gemini | 34.3 | 40.7 | +6.4 |
> | SAM2 & Gemini | 40.5 | 63.2 | +22.7 |
> | VoCap trained on Gemini pseudo-captions | 47.8 | 72.2 | +24.4 |
>
> We will update our paper with the results of the newer Gemini model, to ablate our data pipeline further.
>
> **Comparison to DAM**
>
> The contemporary DAM model (ICCV'25 (October 2025)) performs only image and video captioning. In contrast, our model also performs referring video object segmentation and semi-supervised video object segmentation. DAM cannot easily be extended to these tasks: Its architecture takes an image with a mask and a crop of this image with a mask as input, and generates a caption as output. To caption a video, it first runs the separate SAM2 model to generate masks, which is then fed into the DAM model. In contrast, we have a single model which only requires an input mask on the first frame of a video to generate captions.

---

### Official Review · Reviewer_MS3b · 2025-11-01

**Soundness:** 2
**Presentation:** 2
**Contribution:** 2
**Rating:** 4
**Confidence:** 5

**Summary:**

This paper proposes **VoCap**, a unified framework for video object understanding that can generate spatio-temporal masks and natural language descriptions from any prompt.
The writing is clear and well-structured, with a coherent and novel task formulation supported by a logically developed methodology.
Experimental results demonstrate state-of-the-art performance, confirming the effectiveness and potential of the proposed approach for video segmentation and captioning.

**Strengths:**

The paper explores an interesting direction by attempting to unify video object segmentation and captioning within a single framework.

The idea of leveraging different input modalities (text, box, mask) is conceptually appealing and potentially useful for future multimodal understanding tasks.

The paper is clearly written and easy to follow, with well-organized structure and visual illustrations.

**Weaknesses:**

The proposed VoCap framework mainly stacks existing techniques (SAM2 for segmentation and BLIP2-style text decoding) with minimal methodological innovation.

The model design lacks substantial novelty or clear insight into how segmentation and captioning are effectively integrated beyond simple module combination.

The experimental validation is insufficient and somewhat superficial; it primarily reports improvements on internal benchmarks without solid comparisons to recent or stronger baselines, like DAM[1]

The work does not provide a convincing analysis or explanation to justify the claimed synergy between segmentation and captioning modules.

[1] Lian L, Ding Y, Ge Y, et al. Describe anything: Detailed localized image and video captioning[J]. arXiv preprint arXiv:2504.16072, 2025.

**Questions:**

Missing comparisons with DAM, a relevant and competitive method in multimodal video understanding, undermines the fairness and completeness of the evaluation.

Lacks visual comparisons with other methods.

---

> ### Author Response · Authors · 2025-11-26
> **Please find our response below**
>
> We thank the reviewer for deeming our paper to explore an interesting direction, having experimental results which demonstrate state-of-the-art performance, and for judging our paper to be clearly written and organized.
>
> **Synergy between language and segmentation; Novelty**
>
> Our model combines existing modules into a new framework which we train fully end-to-end. This results in a system which has state-of-the-art performance on referring expression segmentation while also being able to do captioning. Our text tower functions as an encoder when consuming referring expression inputs, and functions as a decoder for captioning. The synergy between segmentation and captioning comes from the fact that the weights are shared (see e.g. Fig.1 in the caption).
>
> Note that our unified architecture allows us to train on captioning and video segmentation tasks jointly. In Table 5, we showed that by pretraining on both captioning and video segmentation tasks (Row 1), we were able to improve performance substantially over only training on video segmentation (Row 3). Concretely, we improved the RefVoS J&F by 3.7 points from 66.6 to 70.3. This result therefore effectively shows the synergies between language and segmentation achieved by our model.
>
> Additionally we introduce a new automatically annotated dataset and a human annotated evaluation dataset on top of SAV. Together we feel our package has sufficient novelty for a top-tier publication.
>
> **Experimental validation and Comparison to DAM**
>
> First of all, we outperform the state of the art on referring expression segmentation on standard and established benchmarks (Table 4). Especially on the highly challenging MeVis dataset, we report large performance gains with respect to very recent baselines like SAMWISE and FindTrack. This convincingly demonstrates that our method is state-of-the-art in referring expression segmentation which is also acknowledged by reviewers UGYr and p7Gk.
>
> The Describe Anything Model (DAM) was published at ICCV’25 (after the ICLR submission deadline) and should be considered contemporary work as per [ICLR policy](https://iclr.cc/Conferences/2026/ReviewerGuide). This paper is indeed relevant and we will incorporate this into our related work. However DAM is quite different to our model. It performs only image and video captioning. In stark contrast, our model also performs referring video object segmentation and semi-supervised video object segmentation. DAM cannot easily be extended to these tasks: Its architecture takes an image with a mask and a crop of this image with a mask as input, and generates a caption as output. This is extended to video by concatenating the visual feature tokens of all its frames. To caption a video, it first runs the separate SAM2 model to generate masks, which is then fed into the DAM model. In contrast, we have a single model which only requires an input mask on the first frame of a video to generate captions.

---

> > ### Comment · Reviewer_MS3b · 2025-11-28
> > **Official comments**
> >
> > Thank you to the authors for addressing my questions. Most of my concerns have been resolved, and I find the idea and implementation of this work quite convincing. Therefore, I decide to increase my score for this submission.
> > But there are still some suggestions.
> > To the best of my knowledge, DAM has been available for quite some time, and I would like to see your model evaluated on the DAM dataset. In addition, for the completeness of the paper, I hope the authors can include ablation studies comparing different backbones.

---

> > > ### Author Response · Authors · 2025-12-03
> > > **Thanks for raising your score**
> > >
> > > We thank the reviewer for signalling their willingness to increase their score to a positive acceptance recommendation.
> > >
> > > As for the request, we re-iterate that DAM is contemporary work and therefore comparisons are not required as per ICLR policy. Nevertheless, we do agree that a comparison would be beneficial to the community. Now the dataset DAM introduced (DLC-Bench) is an image dataset while our model focuses on video. Instead, we are working to compare VoCap to DAM on their video captioning experiments. We will add this to our final manuscript.

---

### Meta-Review · Area_Chair_NxXn · 2026-01-08

**Summary:**

The paper proposes VoCap, a promptable video model that takes a video plus a prompt in multiple modalities (text, bounding box, or mask) and outputs (1) a spatio-temporal object mask sequence (“masklet”) and (2) an object-centric caption describing the segmented target. The goal is to unify promptable video object segmentation, referring expression video segmentation, and video object captioning in a single system


The reviewers generally agree that the paper presents a timely and well-motivated attempt to unify promptable video object segmentation with object-centric video captioning in a single framework. The unified formulation, support for multiple prompt modalities (text/box/mask), and the large-scale SAV-Caption dataset are consistently viewed as strong contributions. Reviewers also find the empirical results on referring expression video object segmentation to be solid, and several appreciate the clarity of the presentation, thoughtful annotation guidelines, and compelling qualitative examples demonstrating long-term object tracking with captions.

At the same time, reviewers raise recurring concerns about limited novelty, noting that the approach largely combines existing promptable segmentation and vision–language components. A major point of discussion is the reliance on pseudo-generated captions, with questions about caption quality, bias, and the strength of supervision despite human validation on the validation set. Multiple reviewers feel that the video object captioning task and evaluation are underdeveloped, with limited baselines and analysis compared to the segmentation results. There is also shared concern that the paper does not sufficiently disentangle whether performance gains stem from architectural choices or from data scale, and that clearer ablations and a more focused experimental breakdown would strengthen the claims.

After considering all the reviews, rebuttal and main paper, the AC decides to reject this work. More comparison experiments should be added due to the novelty issues with full confidence.

**Reviewer Concerns:**

Reviewer: MS3b

Others are still outstanding.

**Reviewer Scores:**

Reviewer: MS3b may raise his score.

Others are not.

---

### Decision · Program_Chairs · 2026-01-26

Reject